# Expression of Nutritional Traits in Vegetable Cowpea Grown under Various South African Agro-Ecological Conditions

**DOI:** 10.3390/plants11111422

**Published:** 2022-05-27

**Authors:** Abe Shegro Gerrano, Ntombokulunga W. Mbuma, Rita H. Mumm

**Affiliations:** 1Agricultural Research Council—Vegetables, Industrial and Medicinal Plants, Private Bag X293, Pretoria 0001, South Africa; 2Food Security and Safety Focus Area, Faculty of Natural and Agricultural Sciences, North-West University, Private Bag X2046, Mmabatho 2790, South Africa; 3Department of Plant Sciences, Faculty of Natural and Agriculture Sciences, University of the Free State, P.O. Box 339, Bloemfontein 9300, South Africa; MbumaNW@ufs.ac.za; 4Department of Crop Sciences and the Illinois Plant Breeding Center, College of Agricultural, Consumer and Environmental Sciences, University of Illinois at Urbana-Champaign, Urbana, IL 61802, USA; ritamumm@illinois.edu; 5African Orphan Crops Consortium, World Agroforestry Centre, P.O. Box 30677, Nairobi 00100, Kenya

**Keywords:** cowpea leaves, vegetable cowpea, genotype by environment interaction, GxE, minerals, protein content, nutrition, genetic improvement

## Abstract

Cowpea (*Vigna unguiculata* L.), a traditional legume food crop indigenous to Africa, has potential as both a vegetable and grain crop in contributing to dietary diversity to support health and address malnutrition, especially for those relying heavily on wheat, maize, and rice. The expression of nutritional traits (protein content and concentrations of iron (Fe), zinc (Zn), and manganese (Mn)) in cowpea leaves was evaluated over diverse agro-ecologies of South Africa and typical agronomic practices of smallholder farmers. The genotypes evaluated displayed genetic variation for all four traits. The mean values of Fe, Zn, Mn and protein content varied from 33.11 to 69.03 mg.100.g^−1^; 4.00 to 4.70 mg.100.g^−1^; and 14.40 to 19.63 mg.100.g^−1^ and 27.98 to 31.98%, respectively. The correlation analysis revealed significant degree of positive association between protein and Zn (*r* = 0.20), while negative associations were observed between Mn and protein (−0.46) and between Mn and Fe (*r* = −0.27). Furthermore, the expression of these important nutrient traits was influenced by the climatic conditions represented by six environments (location by year combinations) as is typical of ‘quality’ traits. Additionally, genotype-by-environment interaction effects were detected, suggesting that local soil properties and soil health may play a role in nutritional content in plants, perhaps particularly for legume crops that rely on symbiotic relationships with soil bacterial populations to fix nitrogen, which is crucial to protein formation. Further studies are needed to understand how to coordinate and align agronomic and soil management practices in vegetable cowpea production, especially those workable for the smallholder farmer, to realize the full genetic potential and nutritional value of improved vegetable cowpea varieties.

## 1. Introduction

Malnutrition is a grave and growing concern, particularly in Africa where an estimated 21% of the population are undernourished [1]. A critical consequence of inadequate nutrition in the first 1000 days of life is stunting, a condition affecting a staggering 58.8 million African children under the age of 5. Malnutrition not only robs human potential but it also comes with significant economic cost to African countries due to reduced productivity in adulthood, lowering per capita GDP (gross domestic product) by an average of 13.5% [2,3]. Among the most serious nutritional deficiencies, protein as well as nutrients including iron (Fe), zinc (Zn), and Vitamin A stand out [4]. Iron deficiency can lead to anemia and impaired mental development and poor pregnancy outcomes. Zinc deficiency is associated with stunting and compromised immune response. Vitamin A deficiency can result in blindness. Other micronutrient minerals such as manganese (Mn) are commonly deficient in African diets as well [4]. Deficiency of Mn can lead to osteoporosis, diabetes, and epilepsy.

Interventions to overcome nutritional deficiencies include supplementing food materials with trace elements; however, this approach is not practical in the long term due to issues in distributing supplements to large populations and associated costs. Therefore, improving the nutritional quality of protein-packed food crops such as legumes (e.g., cowpea, Bambara groundnut, soybean), is considered a viable alternative approach [5].

Cowpea (*Vigna unguiculata* (L.) Walp. 2n = 2x = 22) is an important legume crop indigenous to Africa. Grown mainly in Africa by smallholder farmers for household consumption, cowpea is an important source of protein in African diets [6]. Cowpea offers high yield potential even under low input crop production systems in arid and semi-arid agro-ecologies [7]. The grain is composed of 15 to 25% protein content, 50 to 60% carbohydrate, and 1% fat [8,9,10]. Although grain is the primary focus of the cowpea production for human use, leaves, immature pods, and flowers are also consumed in some parts of the world, especially in Africa [11,12]. Several studies have investigated the yield potential and nutrient content of cowpea as a grain crop [8,9,10,13] and as a vegetable crop utilizing the leaves [11,12,14,15,16,17], as well as immature pods [18]. Considering genetic improvement of vegetable cowpea, previous studies have shown protein content in leaves to range from 29.4 to 33.1% for 23 varieties [14] and from 25.0 to 34.4% for five varieties [16] evaluated in Tanzania. Similarly, cowpea leaf protein content (dry weight basis) of 13 varieties varied from 35.0 to 43.1% [15] in studies conducted in Malawi. Among 20 landraces evaluated in Uganda and Tanzania, protein content in leaves of 21.5 to 40.3% was recorded [17].

Ref. [17] also found Fe concentration and beta-carotene (β) content ranging from 140.5 to 3994.7 µg g^−1^ and 4.1 to 30.5 mg.100.g^−1^, respectively, while [14] recorded values of up to 18.7 mg of Fe, 0.547 mg of Zn, and 4.45 mg of β per 100 g of edible portion of freeze-dried raw cowpea leaves were observed. In South Africa, mineral concentrations in cowpea leaves have been reported to be 142 to 626 mg kg^−1^ for Fe, 49 to 104 mg kg^−1^ for Zn, 196 to 394 mg kg^−1^ for Mn, 8.6 to 19.7 mg kg^−1^ for copper, and 42 to 55 mg kg^−1^ β [11]. These studies have demonstrated that cowpea leaves are a source of high protein and other nutrients contributing to a healthy diet and suggest that breeding could be a mode to further enhance nutritional aspects of cowpea as a vegetable crop. However, the adaptability and stability of cowpea genotypes for quality and nutritional value of fresh leaves has not been comprehensively investigated across different environments (i.e., locations, growing seasons, years, and cultural practices) as the studies mentioned above involved single-location trials and, thus, could not fully explore the effects of Genotype by Environment interaction (GxE), especially those associated with diverse geographies.

Ideally in cultivar development, an improved variety would provide high level performance over a wide range of locations/geographies within a given target market. Significant GxE can limit the scope of coverage for a particular variety that provides outstanding performance in some but not all regions comprising the target market. That is, GxE can lead to a change in rank of lines for a given trait, requiring multiple improved varieties to serve the various environments comprising the target market.

In South Africa, the main cowpea producing provinces are Limpopo, KwaZulu-Natal, Mpumalanga, and North West [19]. These provinces represent a collection of unique environmental conditions with varying climatic characteristics, such as soil type, soil fertility, rainfall, pH, and altitude, that may affect the production and productivity of the crop yield and the nutritional concentration absorbed from the soil into the plant. For example, it has been reported that the symbiotic interaction between rhizobia and their legume hosts are affected by environmental factors such as high temperature, pH, soil nitrogen, drought, and salinity [20]. Hence, the differences across these diverse environments in South Africa could necessitate the need to develop cowpea varieties specifically adapted to these unique environments. Understanding the magnitude and nature of GxE effects will enhance the breeding and selection efficiency in the development of vegetable cowpea varieties that meet farmer and consumer needs and preferences [21,22]. In essence, GxE effects can be exploited as opportunities for genetic improvement customized for specific locales [23]. Recent studies focusing on the composition of legume grains have suggested that grain composition traits vary not only by genotype and by environment, but are also subject to GxE [24,25]. Specifically, the studies of cowpea grain conducted in Africa have highlighted GxE as a factor in expression of nutritional traits [13,26,27,28,29,30,31].

Hence, this present work builds on previous research by [12] to underpin the creation of new varieties of vegetable cowpea that are rich sources of nutrition coupled with outstanding agronomic performance for yield and other traits important to farmers. Twenty-five diverse cowpea genotypes obtained from the Agricultural Research Council (ARC) genebank, South Africa were evaluated to determine the variability and heritability for protein and mineral content [12]. The study revealed ample genetic variability and moderate to high heritability across nutritional traits measured in single location trials. Consequently, a set of 15 high performing lines for various characteristics were identified to further explore genetic potential for nutritional aspects in the development of high-performance vegetable cowpea varieties for South Africa. These 15 genotypes have been evaluated with respect to their potential in developing high yielding grain varieties; notably, GxE was not a factor in expression of grain yield in trial grown at three locations across 2 years [32].

The goal of this work is to assess the genetic control of protein content and selected essential nutrient mineral concentrations of Fe, Zn, and Mn in the set of 15 cowpea genotypes identified as having merit in vegetable cowpea improvement, based on performance over six environments (location by year combinations) representing the cowpea production area of South Africa. Of the nine mineral elements evaluated by [12], this study focused on Fe, Zn, and Mn because they are scarce in most food sources. This work focuses on the effects of Genotype, Environment, and GxE in the expression of these traits. Heritability and correlation among traits were estimated to provide guidance on breeding approaches that could be used in the development of highly productive, highly nutritious vegetable cowpea varieties for South Africa that are sources of high protein and essential micronutrients Fe, Zn, and Mn.

## 2. Materials and Methods

### 2.1. Materials Evaluated

Fifteen cowpea accessions were obtained from the Agricultural Research Council (ARC) genebank collection, with significant overlap with the set of genotypes evaluated for grain yield and agronomic and nutritional traits, where preliminary selection had been carried out by [12] (Table 1). Ten locally adapted South African lines are genotypes evaluated and identified by the ARC as part of its pure line selection from the core collection [32]. These 10 lines represent a subset of the 25 cowpea genotypes that were evaluated for grain yield, protein content, and specific minerals at Roodeplaat Research Station [12,13], which are being used for new population development for the biofortification breeding programme of the ARC. In addition, four Nigerian lines and one Kenyan line considered to be drought tolerant and good sources of protein, Fe, Zn, and Mn were obtained from International Institute of Tropical Agriculture (IITA) and included in the study.

### 2.2. Trial Environments and Their Description

The study was conducted in six environments (location by year combinations) involving four locations in the South African cowpea production areas at Mafikeng, Potchefstroom, Roodeplaat, and Venda during the 2016–2017 summer cropping season, with trials repeated at the Potchefstroom and Roodeplaat in the 2017–2018 summer cropping season. The test sites represented unique environmental conditions for the production of cowpea in the country in terms of geography, soil characteristics including natural fertility, and climate (Table 2).

### 2.3. Experimental Design, Trial Establishment and Management

The field trials were arranged in a randomized complete block design (RCBD) with three replications for each of the six environments. Each genotype was grown in a three-row plot 2 m long with inter-row and in-row spacing of 1 m and 0.50 m, respectively. Two seeds were hand sown per station and slightly wrapped in soil. Trials were irrigated as necessary during the critical stages of germination and seedling emergence to establish a uniform plant stand. In terms of agronomic management practices for the testing sites, no fertilizer was added to the soil to simulate common low-input practices of farmers in the prospective target market; weeds were controlled manually.

### 2.4. Data Collection

Young leaves from 5 randomly selected plants at the middle of the row of each plot were harvested at 6 weeks after planting. The leaves were bright green, soft, and fresh-looking with smooth edges during sampling. The leaf size varies according to its appearance (green, soft, and fresh) during harvesting. The leaves were bulked per plot, oven dried, and ground into fine powder in preparation for analysis. Protein content and the concentrations of Fe, Zn, and Mn were determined at the ARC analytical laboratory. Protein content was estimated using the Kjeldahl method for the quantitative determination of nitrogen described in [33]. This method involves protein digestion, distillation, and determination of percentage (%) nitrogen content of the distillate by titration and then multiplying the % nitrogen by a factor of 6.25 to obtain the corresponding protein content in % [34]. Mineral concentrations of Fe, Zn, and Mn were determined as described in [33]. Approximately 0.5 g of finely ground dried samples was wet digested using a mixture of nitric acid (65%) and hydrochloric acid (37%) (1:3 *v*/*v*). Digestion was conducted on a 95 °C hot plate. Each sample was digested in triplicates. Mineral elements in the digested plant materials were determined using the inductively coupled plasma optical emission spectrometry (ICP-OES).

### 2.5. Data Analysis

Data for protein content and Fe, Zn and Mn concentrations were analysed using linear mixed models of Statistical Analysis System [35]. A combined analysis of variance (ANOVA) across the six environments was implement, with Genotype considered as a fixed effect and Environment considered as random and as representing a sample of all possible cowpea production environments in South Africa. An Expected Means Square (EMS) table was developed to list sources of variation and components contributing to each source of variation in advance of the analysis (Table 3).

Based on components of expected mean squares for each source of variation, tests of significance for Environment, Genotype, and GxE focus on the *F* value, computed as follows:(1)$$F_{E}=\frac{MS_{environ}}{MS_{block/environ}}$$
(2)$$F_{G}=\frac{MS_{genotype}}{MS_{GE}}$$
(3)$$F_{GE}=\frac{MS_{GE}}{MS_{error}}$$

An estimate of genotype variance (*V_G_*) was calculated using the mean square for genotype and the mean square error.
(4)$$V_{G}=\frac{MS_{genotype}-MS_{GE}}{re}$$

An estimate of genotype x environment interaction variance (*V_GE_*) was calculated using the mean square for *GE* and the mean square error.
(5)$$V_{GE}=\frac{MS_{GE}-MS_{error}}{r}$$

An estimate of the phenotypic variance (*V_P_*) was calculated as follows.
(6)$$V_{P}=\frac{V_{error}}{re}+\frac{V_{GE}}{e}+V_{G}$$

Moreover, using *V_G_* and *V_P_*, an estimate of broad sense heritability (*H*^2^) was derived.
(7)$$H^{2}=\frac{V_{G}}{V_{P}}$$

Pearson’s correlation coefficient analysis was performed in SAS using Proc CORR to determine the association of traits for expression of protein, Fe, Zn, and Mn concentrations. Principal component analysis (PCA) was performed to further detail the association among traits. PCA was constructed using GenStat version 20th edition (VSN International, Hempstead, UK) [36].

## 3. Results and Discussion

### 3.1. Analytical Results and Trait Heritabilities

The ANOVAs performed for protein content and concentrations of Fe, Zn, and Mn highlighted the significance of effects of Genotype, Environment, and GxE (Table 4). The effect of Genotype was very highly significant (*p* < 0.001) for protein content, Fe, and Mn and significant (*p* < 0.05) for Zn, demonstrating real differences among the 15 vegetable cowpea entries for expression of these traits and suggesting genetic variability that could be used in varietal improvement. Furthermore, environmental effects for all traits were very highly significant (*p* < 0.001), reflecting the diversity of conditions among the six testing environments. The expression of ‘quality’ traits is typically impacted by climatic conditions, particularly rainfall and temperature. Moreover, the effect of GxE was very highly significant (*p* < 0.001) for protein content, Fe, and Mn and very significant (*p* < 0.01) for Zn, suggesting changes of rank among cowpea entries based on trait means. Clearly, the vegetable cowpea entries respond differently to unique environments.

A comparison of F values representing the effects of Environment, Genotype, and GxE shows the relatively large impact of Environment (Table 4). F values for the test of Environment are 4x–9x larger than F values for Genotype and GxE (whereas in principle, the F values would be 1 if effects were nil). This effect of Environment, which incorporates factors associated with geographical location, climate, and soil conditions, is likely inflated by the wide differences soil properties, which are known to influence plant nutrient content. No fertilizers were added to testing sites in keeping with typical practices of smallholder farmers; results were dependent on natural soil fertility and current soil conditions.

Plant growth and development largely depend on the combination and concentration of mineral nutrients available in the soil. Plants may be challenged in obtaining an adequate supply of these nutrients to meet the demands of basic cellular processes due to their relative immobility. A deficiency of any one of them may result in decreased productivity in the form of biomass or seed yield or reduced plant quality. Symptoms of nutrient deficiency may include stunted growth, death of plant tissue, or yellowing of the leaves due to reduced chlorophyll production, a pigment needed for photosynthesis.

Other studies surveying protein in legumes, particularly cowpea, have emphasized the influence of soil fertility, including nitrogen (N) and phosphorus (P), in the expression of protein content in cowpea [37,38]. Soil N and soil pH are factors that affect the symbiotic relationship between legumes and N-fixing bacteria, which in turn contribute to soil fertility [20]. Still, other studies surveying micronutrient content in legumes, particularly cowpea, emphasize the influence of soil micronutrient availability on Fe and Zn content in the plant [39]. For example, soil organic matter is a critical factor influencing the availability and uptake of soil micronutrients to the plant, with plant micronutrient deficiencies more likely to occur with soils low in organic matter and sandy soils (Washington State University Extension, https://smallgrains.wsu.edu/soil-and-water-resources/essential-nutrients/micronutrients/) (accessed on 4 May 2022). Soil pH also influences the availability of micronutrients for uptake by the plant (University of Illinois Extension, http://extension.cropsciences.illinois.edu/handbook/pdfs/chapter08.pdf) (accessed on 4 May 2022). Clearly, there is an opportunity to integrate agronomics into the development of more nutritious vegetable cowpea varieties to maximize the inherent genetic potential and customize management practices for cowpea growers in the various regions of South Africa.

It is worth noting that GxE was not a factor in expression of grain yield across diverse South African environments with the very same set of 15 genotypes [32]. Implications for genetic improvement, as well agronomic practices, to maximize both yield characteristics affected by mass and nutritional characteristics pertaining to quality point to a requirement for coordinated approaches.

The exaggerated Environment and GxE effects for protein content and Fe, Zn, and Mn concentrations tempered heritability of the traits; H^2^ was estimated at 0.54, 0.12, 0.09, and 0.56, respectively (Table 4) across the six environments. These are much reduced from estimates obtained by [12] for protein content and Fe, Zn, and Mn concentrations at a single location over 2 years, 0.78, 0.99, 0.65, and 0.52, respectively, again pointing to the effect of varying soil conditions as a major underlying cause.

Table 4 also shows that the year effect of weather was not important to the expression of Mn; it was not as much a factor with protein and Mn as other factors of the environment such as soil contributing to the variation effect. Similarly, for Fe and Zn, the weather across the two years did play a significant and more important role in trait expression. However, for both of these traits, other factors of the environment contributed more than 65% of the variation in trait expression. This suggests the effect of soil properties including soil fertility and soil pH played largely into the expression of Protein, Fe, Zn, and Mn, with Protein and Mn affected more for the set of environments tested. Nutritional traits were compared within- and between-environmental zones.

### 3.2. Performance of Cowpea Entries

The means of the 15 cowpea genotypes at each of the six environments (location by year combinations) for protein content and concentrations of Fe, Zn, and Mn are given in the Appendix A, respectively, along with grand means for each environment. Overall means for each genotype for each trait are shown in Table 5, along with the grand mean for each trait.

Across the 15 cowpea genotypes, mean values for protein content ranged from 27.98 to 31.88%, with a grand mean of 29.71% (Table 5). Mean values for Fe concentration ranged from 33.11 to 69.03 mg.100.g^−1^ with a grand mean of 52.20 mg.100.g^−1^. Mean values for Zn concentration ranged from 4.00 to 4.70 mg.100.g^−1^, with a grand mean of 4.26 mg.100.g^−1^. Mean values for Mn ranged from 14.40 to 19.63 mg.100.g^−1^ with a grand mean of 17.08 mg.100.g^−1^. The range of mean values for protein, Fe, and Zn are within the range as reported in other studies [12,40]. However, the range of mean values for Mn are very high compared to those seen in previous studies.

Seven genotypes (Kisumu mix, Mamlaka, Veg cowpea 2, Vigna Onb, VCDC, TVU-14196, and ITOOK-1060) exhibited high protein content, i.e., means above the grand mean (Table 5). Based on the seven genotypes with high protein content (protein is considered as a primary trait), two genotypes (Veg cowpea 2, Meter long bean, 5431, and VCDC) had high Fe concentration, four genotypes (Veg cowpea 2, VCDC, TVU-14196, Meter long bean, 5431, and ITOOK-1060) had high Zn concentration, and only one genotype (Meter long bean) had the highest concentration of Mn compared to grand mean among the test genotypes. Genotype Veg cowpea 2 and VCDC exhibited relatively high protein, Fe, and Zn and identified for multiple traits for nutritional quality breeding. Genotype TVU-14196 exhibited relatively high protein content, Zn, and Mn concentration. The genotype Meter long bean was identified for the concentration of Fe, Zn, and Mn. These results suggest the possibility to successfully breed for high protein content cowpea genotypes coupled with high levels of other important micro-nutrients such as Fe, Mn, and Zn. Proteins are large, complex molecules that play many critical roles in the body. They do most of the work in cells and are required for the structure, function, and regulation of the body’s tissues and organs. On the other hand, micronutrient minerals are needed by the body in small amounts. However, their impact on a body’s health is critical, and deficiency in any of them can cause severe and even life-threatening conditions.

The recommended dietary allowance (RDA) for protein for a healthy adult (≥19 years) is set at 0.8 g of protein per kg of ideal body weight per day. Acceptable macronutrient distribution range for protein is 10 to 35% of energy for adults [41]. The RDA for Fe, Zn and Mn ranged from 13.7 to 20.5 mg, 8 to 11 mg, and 1.8 to 2.3 mg per day, respectively (https://www.news-medical.net/health/Macrominerals-and-Trace-Minerals-in-the-Diet.aspx) (accessed on 4 May 2022).

In light of the RDAs given, one 100 mg serving of cowpea leaves from fifteen genotypes would provide on average about half the recommended amount of protein and about one-third to half of the RDA of Zn, while delivering the full RDA of Fe and Mn (Table 6). These results indicate that cowpea leaves are a good source of essential nutrients and that, included regularly in the diet, vegetable cowpea has value in mitigating problems associated with malnutrition and food security.

### 3.3. Guidance to Future Breeding Efforts

Pearson correlation coefficients were computed for every pair of nutritional traits. Considering protein content as the primary nutritional trait, the lack of negative correlations between protein and both Fe and Zn concentrations bodes well for the potential to simultaneously improve all three traits (Table 7). Fe concentration was not correlated with protein content (*r* = 0.07), whereas Zn concentration was positively correlated with protein content (*r* = 0.20). However, Mn concentration correlated negatively with protein content (*r* = −0.46) and Fe concentration (*r* = −0.27), and it is not correlated with Zn concentration (*r* = −0.06) (Table 7), challenging the prospect for higher nutritional value for all four traits in improved vegetable cowpea varieties.

Directionally, these correlations among pairs of traits are consistent with [12] for the most part, although in that study, Fe concentration was negatively correlated with protein content and positively correlated with Mn concentration. The differences may be due to the fact that only a subset of lines was chosen from the 2015 work to be used in the present study; thus, the composition of the population differs between studies. Furthermore, it should be noted that neither of the correlations mentioned from [12] was statistically significant and may be anomalies.

PCA was performed to account for the variation in data and to reduce its dimensionality to enhance interpretation of the data; PCs are a series of linear least squares fits, with each orthogonal to all previous ones. The sign, either positive or negative, of the variables in each eigenvector is an indication of the direction of the correlation between the components and suggests the positioning of the components within the quadrants of the PCA biplots [42].

PCA results aligned with computed correlations (Table 8). The first principal component, PC1, explained 41.07% of the total variation among the 15 cowpea genotypes for protein content and Fe, Mn, Zn, and Mn concentrations. PC1 featured positive values for Protein content, Fe, and Zn, with Protein being the most heavily weighted; Mn was negatively weighted in keeping with the negative correlations computed in association with all other traits. PC2, accounting for 23.98% of the total variation, placed greatest emphasis on Zn concentration, and PC3, accounting for 23.30% of the total variation, balanced Protein content and Zn while minimizing Fe. PC4 accounted for merely 11.65% of the total variation. The fact that the vast majority of the total variation is not explained by the first two principal components reflects the complicated associations among the traits.

The existence of significant positive correlations found between Zn with protein and Fe suggest that the concentrations for these essential minerals can be improved simultaneously (Table 7). The significant negative correlation observed between protein content with Mn indicated that an improvement of one of these traits will have a negative influence on each other. Similarly, significant negative correlations had been observed between Mn and Fe (Table 7). The use of a selection index in breeding and selection can help in making forward progress with all traits simultaneously. Breeders could consider incorporating nutritional quality traits and yield or other key agronomic traits in the selection index as well to circumvent any negative associations as cultivars possessing all traits preferred by farmers and consumers are developed.

From among the seven genotypes displaying higher-than-average protein content, Kisumu mix and VCDC exhibited high Fe; Veg cowpea 2, VCDC, TVU-14196, Meter long bean, 5431, and ITOOK-1060 exhibited high Zn; Meter long bean and 98K-5301 exhibited the highest concentration of Mn (compared to grand means for these traits) (Table 7). Notably, Veg cowpea 2 and VCDC displayed good performance for protein, Fe, and Zn. These lines could be considered as candidates to initiate breeding efforts for nutritionally improved vegetable cowpea for South Africa and other target markets.

## 4. Conclusions

Significant effects for Genotype, Environment, and GxE were observed among 15 cowpea genotypes for nutritional traits, namely, protein content as well as Fe, Zn, and Mn concentrations. Several accessions from the ARC genebank were identified as potential parents to initiate or enhance breeding efforts to develop nutritionally improved vegetable cowpea varieties. As for the genetic control of key nutrient traits, low to moderate heritability estimates obtained for all four traits indicated that the proportion of phenotypic variation attributable to genetics was low compared to variation introduced through the environment and error. Heritability estimates differed dramatically from those obtained for the same traits in earlier studies by [12] based on a single-location trial. The findings suggested that effects of environment and GxE could be attributable to differences in soil properties represented by the four locations comprising the test environment’s bearing on the availability of nutrients from the soil for uptake and use by the plant, and ultimately for nutritional value in the human diet. Further study is needed to understand how to coordinate and align agronomic practices and soil management in vegetable cowpea production, especially those workable for the smallholder farmer, to realize the full genetic potential and nutritional value of improved vegetable cowpea varieties.

## Figures and Tables

**Table 1 plants-11-01422-t001:** Description of cowpea genotypes.

Genotype	Origin	Growth Habit	Notable Characteristics
Veg cowpea 1	South Africa	Semi-upright	Grain yield and related traits
TVU-14196	Nigeria	Semi-upright	Protein, Fe, Zn, and Mn content in the immature pods
Veg cowpea 2	South Africa	Semi-upright	Protein, Fe, Zn, and Mn content in the fresh leaves
Meter long bean	South Africa	Prostrate	Protein, Fe, Zn, and Mn content in the fresh leaves
Vigna Onb	South Africa	Prostrate	Protein, Fe, Zn, and Mn content in the fresh leaves
Kisumu mix	Kenya	Prostrate	Protein, Fe, Zn, and Mn content in the fresh leaves
M217	South Africa	Upright	Protein, Fe, Zn, and Mn content in the fresh leaves
Ukaluleni	South Africa	Prostrate	Grain yield and agronomic traits
VCDC *	South Africa	Upright	Grain yield and agronomic traits
5431	South Africa	Upright	Grain yield and agronomic traits
Chappy	South Africa	Prostrate	Grain yield and agronomic traits
Mamlaka	South Africa	Semi-upright	Protein, Fe, Zn, and Mn content in the immature pods
IT96D-602	Nigeria	Upright	Protein, Fe, Zn, and Mn content in the immature pods
98K-5301	Nigeria	Upright	Protein, Fe, Zn, and Mn content in the immature pods
ITOOK-1060	Nigeria	Upright	Protein, Fe, Zn, and Mn content in the immature pods

* VCDC = Vegetable cowpea Dakama Cream.

**Table 2 plants-11-01422-t002:** Environmental characteristics of trial sites.

Environments	Year	Soil Type	Soil pH	Soil Nutrients (mg kg^−1^)	Fertiliser Applied	Altitude (m.a.sl)	Total Annual Rainfall (mm)	Average Annual Temperature (°C)	Latitude	Longitude
Mafikeng	2016	Red sandy loam	7.1	P (11), K (290), Ca (390), Mg (163), Na (5), N (376), C (0.37), S (4)	None	1369	730.0	23.8	25°85’ N	25°64’ E
Potchefstroom (Potch)	2016	Sandy clay loam	5.8	N-NO_3_ (2.25), N-NH_4_ (1.5), P (41), K (248)	None	1340	666.0	23.9	26°71’ S	27°09’ E
Potchefstroom (Potch)	2017	Sandy clay loam	5.8	N-NO_3_ (2.25), N-NH_4_ (1.5), P (41), K (248)	None	1340	542.0	22.3	26°71’ S	27°09’ E
Roodeplaat (RPT)	2016	Clay loam	7.1	P (92), K (147), Ca (1413), Mg (548), Na (62), Zn (8.97), Fe (38.0), Cu (6.72), Mn (98.4), S (139.2), exchangeable cation Ca (%) = 60.2, exchangeable cation Mg (%) = 29.2, exchangeable cation K (%) = 8.5, exchangeable cation Na (%) = 2.2	None	1168	772.4	24.4	17°49’ S	31°04’ E
Roodeplaat (RPT)	2017	Clay loam	7.1	P (92), K (147), Ca (1413), Mg (548), Na (62), Zn (8.97), Fe (38.0), Cu (6.72), Mn (98.4), S (139.2), exchangeable cation Ca (%) = 60.2, exchangeable cation Mg (%) = 29.2, exchangeable cation K (%) = 8.5, exchangeable cation Na (%) = 2.2	None	1168	711.2	23.9	17°49’ S	31°04’ E
Venda	2016	Red loam	6.06	N (0.19), P (10.1), K (311.0), Na (0.12 cmol kg^−1^), Fe (126.7), Mg (2.5 cmol kg^−1^), Ca (5.8 cmol kg^−1^), (CEC, 23.04)	None	2126	400.0	16.3	23°05’ S	29°49’ E

m.a.sl = meter above sea level; mm = millimeter.

**Table 3 plants-11-01422-t003:** Expected mean square (EMS) table listing sources of variation (SOV), associated degrees of freedom (DF), and components of the variation for the analysis of nutrient traits in cowpea leaves.

SOV	DF	*MS*	EMS
Environment (E)	e − 1	*MS* * _environ_ *	*V_error_* + *gV_B/E_* + *rg**V_environ_*
Block/E	e(r − 1)	*MS* * _block/environ_ *	*V_error_* + *gV_B/E_*
Genotype (G)	g − 1	*MS* * _genotype_ *	*V_error_* + *rV_GE_* + *re**V_genotype_*
GE	(g − 1)(e − 1)	*MS* * _GE_ *	*V_error_* + *rV_GE_*
Error	(g − 1)(r − 1)e	*MS* * _error_ *	*V_error_*
Total	gre − 1		

SOV = sources of variation; DF = degree of freedom; *MS* = mean squares; EMS = error mean squares.

**Table 4 plants-11-01422-t004:** Mean squares, degrees of freedom (df), and F values produced in the analysis of variance as well as estimates of variance for Genotype (G), Genotype by Environment interaction (GxE), and phenotype (P) and broad sense heritability (H^2^) for protein content, and concentrations of Fe (iron), Zn (zinc), and Mn (manganese).

Source of Variation	df	Protein (%)	Fe (mg.100.g^−1^)	Zn (mg.100.g^−1^)	Mn (mg.100.g^−1^)
Environment	5	660.38 ***	37,690.37 ***	76.33 ***	743.66 ***
Block (Env)	12	17.39 ***	1013.99 *	9.91 ***	26.34 ***
Genotype	14	23.33 ***	1973.21 ***	0.65 *	36.21 ***
Genotype × Environment	70	10.78 ***	1733.12 ***	0.59 **	16.00 ***
2016 vs. 2017		142.31 ***	57,062.53 ***	132.66 ***	6.72
Error	168	5.62	475.45	0.34	7.54
Total	269	5535.30	429,438.73	608.17	6928.47
F_E_	5	37.89 ***	37.17 ***	7.70 **	28.25 ***
F_2016 vs. 2017_		25.34 ***	120 ***	391.24 ***	0.89
F_G_	14	4.15 ***	4.15 ***	1.92 *	4.80 ***
F_GxE_	70	1.92 ***	3.65 ***	1.75 **	2.12 ***
σ^2^_G_		0.697	13.338	0.003	1.123
σ^2^_GxE_		1.72	419.223	0.083	2.820
σ^2^_P_		1.296	109.623	0.036	2.012
H^2^		0.54	0.12	0.09	0.56

*** Significant at *p* < 0.001, ** significant at *p* < 0.01, * significant at *p* < 0.05.

**Table 5 plants-11-01422-t005:** Least square means across all environments for each cowpea genotype as well as Least Significant Differences (LSDs), coefficients of variation (CVs), and the grand mean (GM) for each trait.

Genotype	Protein (%)	Fe (mg.100.g^−1^)	Zn (mg.100.g^−1^)	Mn (mg.100.g^−1^)
Veg cowpea 1	28.92	54.78	4.05	18.45
TVU-14196	29.74	48.63	4.39	18.01
Veg cowpea 2	31.12	60.94	4.35	16.71
Meter long bean	27.98	68.04	4.70	19.63
Vigna Onb	30.71	49.65	4.20	16.84
Kisumu mix	31.88	33.11	4.14	14.40
M217	28.68	59.97	4.00	17.51
Ukaluleni	28.91	44.51	4.07	16.49
VCDC *	30.44	69.03	4.55	16.34
5431	28.72	63.32	4.30	18.42
Chappy	29.41	54.58	4.15	16.56
Mamlaka	31.22	41.18	4.22	17.22
IT96D-602	28.60	46.45	4.25	15.06
98K-5301	29.55	46.49	4.19	18.70
ITOOK-1060	29.73	42.33	4.39	15.78
LSD	2.04	14.30	1.45	2.41
CV (%)	7.98	41.77	13.66	16.08
GM	29.71	52.20	4.26	17.08

* VCDC = Vegetable cowpea Dakama Cream.

**Table 6 plants-11-01422-t006:** Grand means, standard deviation (sd), and range for nutrient traits compared with the Recommended Daily Allowance (RDA) and average nutritional delivery in a 100 g serving of cowpea leaves and correlation of iron (Fe), zinc (Zn), and manganese (Mn) with protein content.

Trait	Mean ± sd	Range	RDA#	Content wrt RDA
Protein	29.71 ± 2.37	27.98–31.22	46–56 g	One 100 g serving of cowpea leaves delivers about half of the RDA for protein
Fe	52.20 ± 21.80	33.11–68.04	8 mg (man), 18 mg (woman), 27 mg pregnant women	All genotypes were above RDA; One serving of cowpea leaves delivers all of the RDA for Fe
Zn	4.26 ± 0.58	4.00–4.70	8.0 (woman)–11.0 (man) mg	All genotypes were below RDA; One serving of cowpea leaves delivers about half of the RDA for Zn
Mn	17.08 ± 2.75	14.40–19.63	1.8 mg (woman)–2.3 mg (man)	All genotypes were above RDA; One serving of cowpea leaves delivers all of the RDA for Mn

**Table 7 plants-11-01422-t007:** Pearson coefficients of phenotypic correlation among nutrient traits: protein content, iron (Fe), zinc (Zn) concentrations, and manganese (Mn) concentrations.

Traits	Protein	Fe	Zn	Mn
Protein	1			
Fe	0.07	1		
Zn	0.20 ***	0.14 *	1	
Mn	−0.46 ***	−0.27 ***	−0.06	1

*** Significant at the *p* < 0.001, * significant at *p* < 0.05.

**Table 8 plants-11-01422-t008:** Eigenvectors from principal component (PC) analysis for cowpea genotypes on protein, iron (Fe), zinc (Zn), and manganese (Mn) content.

Traits	PC1	PC2	PC3	PC4
Protein	0.59	−0.29	0.43	0.62
Fe	0.40	0.36	−0.78	0.33
Zn	0.33	0.80	0.43	−0.25
Mn	−0.62	0.39	0.14	0.67
Eigenvalue	1.64	0.96	0.93	0.47
% Variation	41.07	23.98	23.30	11.65

## Data Availability

Not applicable.

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
