# Peer review of "Expression of Nutritional Traits in Vegetable Cowpea Grown under Various South African Agro-Ecological Conditions"

_plants, 2022, doi:10.3390/plants11111422_

Round 1

Reviewer 1 Report

Dear authors,

I think this study has done a lot of work and effort. Although there is no advanced technique, the data is still valuable for future breeding work, especially in the Special Issue of Vegetables Breeding in South Africa. However, there is still room for improvement and might need to be revised. Please find my suggestions below:

  1. Regarding the abstract part, I feel that the description was too general. Would you emphasize the key findings of the article? Since you have analyzed the content of protein, Fe, Zn, and Mn, you might put the result data in the abstract, too.
  2. In line 72, no subject was found in the sentence. I will also suggest that “beta-carotene” can use the Greek alphabet β.  
  3. For the introduction, there are too many paragraphs. You have 12 paragraphs and some of them are very short, such as in 63-65. There is only one sentence for a paragraph which is weird to me. Actually, some paragraphs can be combined and some of them might be put in the discussion. You might also directly point out the key and importance of the article.
  4. In lines 136-137, with the set of genotypes evaluated by “what”? I also don’t understand clearly regarding “notable characteristics” in the Table 1.
  5. I would expect to have one more section for a description of protein, Fe, Zn, and Mn analysis. It is too simplified in lines 169-173.
  6. In line 166, I expect to understand more about the young leaves that you collected for analysis. Size? Appearance? Maturity?
  7. The description in the result and discussion is quite clear, but I feel a bit lengthy. It will be great if you can refine it. Some format issues can be found. In line 266, 2 should be superscript (H2).
  8. In general, the quality and content of this article are good, but a minor revision is needed.

Author Response

Dear Editor,

Please find attached response cover letter for reviewer 1. The authors would like to thank a reviewer for positive and constructive suggestions.

Regards,

Abe Gerrano (corresponding author)

Reviewer 2 Report

Cowpea,a traditional food crop indigenous to Africa, has potential in contributing to dietary diversity as vegetable and grain food. This study worked on the evaluation of the expression of nutritional traits including protein content and concentrations of iron, zinc, and manganese in cowpea leaves in diverse agro-ecologies of South Africa. The genotypes and environments and genotype-by-environment interaction effects were evaluated and achieved meaningful results for the smallholder farmer to realize the full genetic potential and nutritional value of improved vegetable cowpea varieties.

my major concern: The study has investigated the four nutritional traits, one of which is the protein content that related closely to the culture condition and the yield. The MS maybe need to include the yield in the content and make some comparison with protein content.

The other three investigated nutritional traits are concentrations of iron, zinc, and manganese in cowpea leaves that should closely related to the concentrations of iron, zinc, and manganese in the soil, it is necessary to calculate the coefficient of the concentration of leaves and soils to make clear the concentration of leaves come from soil or from genetics.

There are minor concerns;

  1. There are multiple expressions as ‘genotypes evaluated by [12]’ in P1 L137. I suggest to indicate the method with name instead of the only reference.
  2. There are some mistakes in the context, such as P10 L325 ‘one 100 mg serving of cowpea leaves from 15 genotypes‘,I guess the 100mg should be 100g. P12 table 6 ‘range for nutrient traits compared with with’ , there maybe have double ‘with’.

Author Response

Dear editor,

Please find attached response letter for the reviewer 2. We appreciate a reviewer for the positive and constructive suggestions and comments.

Regards,

Abe Gerrano (corresponding author)

Reviewer 3 Report

Dear authors,

The manuscript is well written, interesting and important for the scientific and professional community.

After reviewing the text, I found a few potential errors, so I will now state where they are in the text:

In the chapter Materials and Methods, line 143, there is the abbreviation IITA. What is behind the abbreviation?

Further, in the chapter Results and Discussion, the data in lines 291 to 294 are not the same as in Table 5. Namely, I don’t think you provided good data for the minimum and maximum values for proteins and Fe.

Also, please check the data you provide in lines 305 to 314, which also refer to the data in Table 5.

Check the data you present in lines 373 to 379, which refer to Table 5.

In Table 6, check the value ranges for proteins and Fe.

You cite 42 authors in the text and there are 43 citations in the references.

After you check and change what I specified the manuscript can be published.

Author Response

Dear editor,

Please find attached response letter for a reviewer 3. We appreciate the reviewer for positive and constructive suggestions and comments.

Regards,

Abe Gerrano (corresponding author)

Round 2

Reviewer 2 Report

The MS has been seriously revised and the questions have been nicely answered.